# Trends in female-selective abortion among Asian diasporas in the United States, United Kingdom, Canada and Australia

Catherine Meh, Prabhat Jha*

Centre for Global Health Research, Unity Health Toronto and Dalla Lana School of Public Health, University of Toronto, Toronto, Canada

**Abstract** Preference for sons and smaller families and, in the case of China, a one-child policy, have contributed to missing girl births in India and China over the last few decades due to sex-selective abortions. Selective abortion occurs also among Indian and Chinese diaspora, but their variability and trends over time are unknown. We examined conditional sex ratio (CSR) of girl births per 1000 boy births among second or third births following earlier daughters or sons in India, China, and their diaspora in Australia, Canada, United Kingdom (UK), and United States (US) drawing upon 18.4 million birth records from census and nationally representative surveys from 1999 to 2019. Among Indian women, the CSR in 2016 for second births following a first daughter favoured boys in India (866), similar to those in diaspora in Australia (888) and Canada (882). For third births following two earlier daughters in 2016, CSRs favoured sons in Canada (520) and Australia (653) even more than in India (769). Among women in China outside the one-child restriction, CSRs in 2015 for second order births somewhat favoured more girls after a first son (1154) but more heavily favoured boys after a first daughter (561). Third-birth CSRs generally fell over time among diaspora, except among Chinese diaspora in the UK and US. In the UK, third-birth CSRs fell among Indian but not among other South Asian diasporas. Selective abortion of girls is notable among Indian diaspora, particularly at higher-order births.

*For correspondence:
Prabhat.jha@utoronto.ca

## Editor's evaluation

This paper provides fundamental epidemiologic evidence on the abnormally skewed sex ratio at birth in Asia with expanded information on the Asian diaspora in OECD countries. India and China are two of the largest populations in the world that are most affected by a long history of strong son preference, aided by access to technology to detect the sex of the fetus and access to selective sex abortion. Both issues are important for maternal health, but also at the individual and family level in terms of kinship structure and the gender balance for future generations. The authors make a compelling case of skewed sex ratios favoring boys that persist among the Asian diaspora even under different social, cultural, or economic norms and constraints.

## Introduction

From 1970 to 2010, an estimated 105 million females (births and all age groups) were missing in China (62 million) and India (43 million), the world's two most populous countries (**Bongaarts and Guilmoto, 2015**). China and India account for 90% of the annual 1.2–1.5 million missing female births globally (**Bongaarts and Guilmoto, 2015**; **Chao et al., 2019**; **UNFPA, 2020**). Until the 1980s, the main driver

of missing females was an excess of female deaths attributable to infanticide and negligence towards female children (*Kumar and Sinha, 2019*). Since about 1985, ultrasound-enabled prenatal sex determination followed by selective abortion of female fetuses has become the main method for families to enact strong existing cultural preferences for sons (*Bongaarts and Guilmoto, 2015*; *Tabaie, 2017*). Selective abortion of female fetuses is driven mostly by son preference but also, in the case of China, the imposition of a 'one-child' policy for most of China's population which began in 1980 and officially ended only in 2016 (*Ebenstein, 2010*; *Feng et al., 2016*).

Examining sex ratio patterns among the diaspora sheds light on the differences between Indian and Chinese practices inside and outside the home countries. There are an estimated 18 million Indian and 10 million Chinese male and female diaspora as of 2020 (*United Nations Population Division, 2021*). Much of the migration from India and China has been to Australia, Canada, the United Kingdom (UK), and the United States (US).

Within India, selective abortion is uncommon with first births. However, a significant minority of families with a first daughter will turn to the practice so as to secure at least one son (*Jha et al., 2006*; *Jha et al., 2011*; *Saikia et al., 2021*). In China, a limited number of ethnic minorities and rural areas were granted exemption to the one-child policy means, which allows examination of selective abortion for the fewer second-order births. To quantify the impacts of selective abortion, we apply the conditional sex ratio (CSR) method, which examines the variation in second or third births in relation to the sex of the earlier child(ren) (*Jha et al., 2011*; *Saikia et al., 2021*). The sex of subsequent children is independent of the first and biological factors, which may affect overall sex ratio, should not vary by birth order (*Jha et al., 2006*). In the absence of sex selection, there is a slight deficit of girls compared to boys (i.e., a CSR of <1000) for biological reasons, leading to a 'natural' sex ratio of 950–975 girls per 1000 boys, a ratio observed consistently over decades in societies where selective abortion is rare (*Chao et al., 2019*; *Guilmoto, 2007*; *Hesketh and Xing, 2006*).

Here, we quantify the trends in CSRs among Asian diaspora populations, particularly among Indians and Chinese. We evaluated CSR trends for Indians and Chinese in home countries and among the diasporas in Australia, Canada, the UK, and the US using census data and representative samples of birth histories over two decades from 1999 to 2019.

# Materials and methods
## Data source
We used multiple rounds of household surveys with birth histories for local families in India and China. For Indian and Chinese diaspora populations in Australia, Canada, the UK, and the US, we relied on census data with information on household composition and member identifiers to reconstruct families and birth histories. We also explored sex ratios in diaspora communities from Bangladesh, Pakistan, Hong Kong, Sri Lanka, and the Republic of Korea.

We extracted complete birth histories from three rounds of the Indian National Family Health Survey (NFHS; 1998–99, 2005–06, and 2015–16), a large periodic nationally representative cross-sectional survey, which provides information on population, health, and nutrition of Indian households. A combined 914,374 women aged 15–49 years were interviewed in the three survey rounds. These surveys use a multistep sample design based on the Indian census-sampling frame and have a 95% average response rate from eligible women. Sampling design and methodology for the NFHS have been published (*International Institute for Population Sciences, 2000*; *International Institute for Population Sciences, 2007*; *International Institute for Population Sciences, 2017*).

The China Health and Nutrition Survey (CHNS) was used to explore sex ratios in China. This is an ongoing survey of select provinces (Beijing, Liaoning, Heilongjiang, Shanghai, Jiangsu, Shandong, Henan, Hubei, Hunan, Guangxi, Guizhou, and Chongqing) and autonomous cities/districts in China, which includes questions on health, nutrition, and family planning (China Center for Disease Control and Prevention, https://www.cpc.unc.edu/projects/china [retrieved November 29, 2021], *Popkin et al., 2010*). The survey collects birth history information from women aged 52 years and younger using a multistage randomized sample design; more than 40,000 individuals have participated in the survey. The CHNS cohort reflects China's age, gender, and education profile in the national census of 2010 (*Zhang et al., 2015*).

Australia and Canada conduct censuses every 5 years, of which we included four for Australia (2001, 2006, 2011, and 2016) and two for Canada (2001 and 2016, excluding the 2011 census which was adversely affected by political interference). Complete (100%) census data for Asian women were available for Australia. The Canadian census captured details on births in the diaspora population using the long-form questionnaire, a more detailed survey of a fraction of the population. In the 2001 and 2016 censuses, 20% and 25% of randomly selected Canadian households completed the long form, respectively (*Statistics Canada, 2016*; *Statistics Canada, 2018*).

We included complete count data from two decennial censuses for the UK (2001 and 2011) covering the population of England and Wales (Scotland and Northern Ireland were excluded). We used the 5% random sample of the US 2000 census and multiple rounds of the 1% American Community Survey (ACS) from 2001 to 2019 pooled by 5-year periods. The ACS is an ongoing annual survey, which provides data on the US population using the census framework and sample units (*US Census Bureau, 2021*).

## Study population

For India and China, the NFHS and CHNS had direct birth histories requiring minimal data manipulation to derive birth order. For the diaspora populations, eligible participants were families with children (male and female) aged 14 years or younger born in Australia, Canada, UK, and US to mothers of Asian ethnicities from India, China, Bangladesh, Pakistan, Hong Kong, South Korea, and Sri Lanka. Mothers were born either in the Asian country of origin (first generation) or in the country of residence (second generation). We identified the latter by first responses to the census question on ethnic origins and linked these to the corresponding Asian country of origin. We stratified first generation mothers by birth country and ethnicity to tease out women born in the Asian country of interest but with nonlocal ethnicity. We included eligible families of the local population, identified as 'domestic', for comparisons within each country. The selected families had four or fewer children in the same household as their female parent. We reconstructed the birth sequence by gender for children born in Western countries.

We excluded families with adopted children, multifetal births (same or dual sex twins, triplets, and higher-order multiples). We also omitted families with five or more children, as sex selection was unlikely. Families that desire a son, without sex selection, continue childbearing and grow large. Those with the added need for a small family size are most likely to resort to prenatal sex selection (*Hesketh and Xing, 2006*). Other exclusions were same sex couples, single father homes, and families with children older than 14 years, to reduce the chance of missing older children away from the household. There was no distinction between adopted and biological children in the Canadian census. However, this is low among the families of interest given the <2% overall adoption rate in Canada (*United Nations Population Division, 2009*). Data for Korean women in the UK were available only in the 2011 census.

## Conditional sex ratio

We defined the natural sex ratio range as 950–975 girls per 1000 boys (*Jha et al., 2011*; *Saikia et al., 2021*). Sex ratio is the total number of female births per 1000 male births (Pf/[1−Pf]) × 1000; Pf is the proportion of females to total births. This ratio also corresponds to 1.05 male per female (*Chao et al., 2019*; *Guilmoto, 2007*), which highlights an excess of male births. We prefer using the ratio of girls per 1000 boys as in earlier analyses (*Jha et al., 2006*; *Jha et al., 2011*; *Saikia et al., 2021*) as we believe it better captures missing girls, the focus of our study.

Declining fertility rates, and smaller family sizes with increased proportions of first borns, affect the overall sex ratio. Therefore, we used CSR, a variant of the sex ratio measure, which takes the sex of previous births into consideration. The CSR reveals further pockets of sex selection in higher-order births, particularly in second and third births with an earlier daughter or two earlier daughters, that are not evident in the overall sex ratio, particularly with falling overall fertility (*Jha et al., 2006*; *Jha et al., 2011*; *Saikia et al., 2021*).

We derived CSRs using a minimum total of 100 births or more for each birth order and its corresponding sex composition of previous births within in each country. We used (n=100) as the minimum for the denominator to ensure stable CSR estimates. All data were analysed using Stata 16.1.

## Results

The study included 18.4 million children aged 14 and below. Of these, 1.03 million were births in India and China. Most births in China were first (65%) and second (28%) born children with few third order births (5%) in all survey years (*Table 1*). In India, about 34% and 29% of all births were first and second births, respectively, and 17% were third born. About 1.1 million were births within the Indian (663,397) and Chinese (429,702) diasporas in Australia, Canada, the UK, and the US, with similar proportionate distributions for first (India 57%; China 60%), second (36%; 34%), and third order births (7%; 6%). Asian diasporas from Bangladesh, Pakistan, Hong Kong, Sri Lanka, and South Korea added 837,234 total births with 50% first, and 35% second born children. Overall, 55% of all study births were first born children and 39% were second born or third born, among whom the effects of conditional sex selection can be best observed. All CSR estimates for births in this study are shown in *Supplementary file 1A-1E*.

### Mothers born in India

For women born in India and giving birth in India or in Canada, Australia, the UK, or the US, the sex ratio for first births remained stable within the natural range (950–975) excepting minor deviations in India in 1999 and 2016, where the CSRs favoured more boys, at 932 and 931 girls per 1000 boys, respectively (*Figure 1*; Panel A). CSRs for second order births with an earlier son were also close to the natural range with a slight upward trend in India from 953 (1999) to 986 (2016). For the Indian diaspora, CSRs were mostly within the natural range for second order births with an earlier son except for the US where they fluctuated (*Supplementary file 1D*).

By contrast, for second and third order births with an earlier daughter or daughters, the CSRs favoured boys, being significantly below the natural range for births in all countries. For second births with an earlier daughter, the Indian diaspora in Canada had the lowest CSR (859 girls per 1000 boys) in 2001. By 2016, CSRs were largely unchanged in Australia (888), Canada (882), and India (866). For the UK and the US, CSRs were 923 (2011) and 898 in (2019), respectively.

The CSRs for third order births with two previous daughters deviated even further from the natural range in all countries and were lower in the diaspora than in India (769) itself. Canada had the lowest CSRs (540–520 girls per 1000 boys) between 2001 and 2016 for this birth order and between 2011 and 2016 in Australia, CSRs declined from 770 to 653 girls per 1000 boys. The 2019 figure from the US suggested the beginning of a possible trend towards the normal range but was based on relatively small numbers of births (*Table 2*).

### Mothers born in China

The CSRs for first born children among families in China and the Chinese diaspora converged just below the natural sex ratio range across all countries, and within China, rose steadily from below the natural range in 1999 to be within the natural range by 2011. Compared to births in the Chinese diaspora, CSRs in China were lowest (864) and were highest in the US (950) in the early 2000s (*Figure 1*, Panel B). Focusing on the diaspora, there were small deviations from the natural range for second order births with one earlier son or daughter, with slight upward and downward trends.

In China itself, however, CSRs were considerably skewed for second order births, which are permitted in rural areas and select populations exempt from the one-child policy (*Guilmoto, 2015*; *Ouyang, 2013*). When the first child was a son, the CSR modestly favoured girls, rising from 1138 girls per 1000 boys in 2000 to 1341 in 2011, before declining to 1154 by 2015 – all well above the natural range. Conversely, when the first child was a daughter, the CSRs strongly favoured boys, falling, from 677 in 2000 to 368 in 2006 and rising to 561 in 2015. These suggest that selective abortion of second males following a first boy was occurring, and to an even greater degree of selective abortion of second girls following a first girl.

There were too few third order births in China for any reliable study. Among third births in the Chinese diaspora, CSRs favoured boys, being below the natural range. CSRs for third births with two earlier daughters increased in the UK from 783 in 2001 to 966 in 2011, and in Canada, it decreased from 903 to 877 girls per 1000 boys. CSRs were lowest in the US ranging from 761 in 2000 to 782 in 2019. In Australia, CSRs increased from 846 in 2001 to 894 in 2011, before dropping to 793 in 2016.

**Table 1.** Number of births by country and birth order.

| Country (year) | Source | Year | First born | Second born, one earlier son | Second born, one earlier daughter | Third born, two earlier sons | Third born, two earlier daughters | Third born, one earlier son and daughter | All births, children 14 and below | Total Births among Indians diaspora | Total Births among Chinese diaspora |
|---|---|---|---|---|---|---|---|---|---|---|---|
| **Births in diaspora settings** | | | | | | | | | | | |
| Australia | Census | 2001 | 530,793 | 176,934 | 168,100 | 32,223 | 28,342 | 47,651 | 1,004,114 | 14,155 | 30,233 |
| Australia | Census | 2006 | 575,914 | 188,474 | 179,022 | 32,211 | 27,898 | 48,020 | 1,071,050 | 19,202 | 37,305 |
| Australia | Census | 2011 | 572,686 | 185,248 | 176,499 | 31,242 | 27,337 | 47,063 | 1,059,141 | 38,649 | 48,175 |
| Australia | Census | 2016 | 617,168 | 200,965 | 191,349 | 31,609 | 27,788 | 47,853 | 1,134,915 | 84,452 | 78,827 |
| Canada | Census | 2001 | 87,718 | 27,882 | 26,404 | 4,345 | 4,042 | NA | 162,082 | 6,639 | 5,931 |
| Canada | Census | 2016 | 958,730 | 295,240 | 279,605 | 38,185 | 35,625 | 60,345 | 1,689,395 | 154,400 | 130,730 |
| United Kingdom | Census | 2001 | 2,926,117 | 918,845 | 866,111 | 138,203 | 118,736 | 203,188 | 5,171,200 | 111,176 | 7,505 |
| United Kingdom | Census | 2011 | 3,256,113 | 944,567 | 892,899 | 133,190 | 116,264 | 204,393 | 5,547,426 | 156,836 | 24,200 |
| United States | Census 5% | 2000 | 35,391 | 13,338 | 12,811 | 2,181 | 2,384 | 3,542 | 72,082 | 9,422 | 9,246 |
| United States | ACS | 2004 | 20,377 | 7,828 | 7,378 | 1,286 | 1,276 | 2,042 | 41,344 | 4,178 | 4,120 |
| United States | ACS | 2009 | 66,217 | 25,696 | 24,235 | 4,260 | 4,106 | 6,615 | 135,263 | 16,887 | 14,588 |
| United States | ACS | 2014 | 78,804 | 31,029 | 29,203 | 5,386 | 5,070 | 8,470 | 164,040 | 20,969 | 17,629 |
| United States | ACS | 2019 | 76,155 | 29,878 | 28,599 | 4,925 | 4,775 | 7,761 | 158,062 | 26,432 | 21,213 |
| Total | | | 9,802,183 | 3,045,924 | 2,882,215 | 459,246 | 403,643 | 686,943 | 17,410,114 | 663,397 | 429,702 |
| Proportion | | | 56% | 17% | 17% | 3% | 2% | 4% | 100% | | |

*Table 1 continued on next page*

*Table 1 continued*

| Country (year) | Source | Year | First born | Second born, one earlier son | Second born, one earlier daughter | Third born, two earlier sons | Third born, two earlier daughters | Third born, one earlier son and daughter | All births, children 14 and below | Total Births among Indians diaspora | Total Births among Chinese diaspora |
|---|---|---|---|---|---|---|---|---|---|---|---|
| **Births in home country** | | | | | | | | | | | |
| China | CHNS | 2000 | 1,687 | 402 | 468 | 59 | 110 | 105 | 2,944 | | |
| China | CHNS | 2006 | 1,325 | 195 | 301 | 16 | 36 | 29 | 1,933 | | |
| China | CHNS | 2011 | 1,561 | 206 | 344 | 8 | 34 | 22 | 2,193 | | |
| China | CHNS | 2015 | 1,364 | 265 | 373 | 7 | 48 | 22 | 2,093 | | |
| India | NFHS-2 | 1998/99 | 44,907 | 20,530 | 19,415 | 7,141 | 7,657 | 14,397 | 161,523 | | |
| India | NFHS-3 | 2005/06 | 48,299 | 21,761 | 21,459 | 6,781 | 8,080 | 14,398 | 166,861 | | |
| India | NFHS-4 | 2015/16 | 250,233 | 106,712 | 104,693 | 24,004 | 35,111 | 53,058 | 693,227 | | |
| Total | | | 349,376 | 150,071 | 147,053 | 38,016 | 51,076 | 82,031 | 1,030,774 | | |
| Proportion | | | 34% | 15% | 14% | 4% | 5% | 8% | 100% | | |
| **All study births** | | | | | | | | | | | |
| Total | | | 10,151,559 | 3,195,995 | 3,029,268 | 497,262 | 454,719 | 768,974 | 18,440,888 | 663,397 | 429,702 |
| Proportion | | | 55% | 17% | 16% | 3% | 2% | 4% | 100% | | |

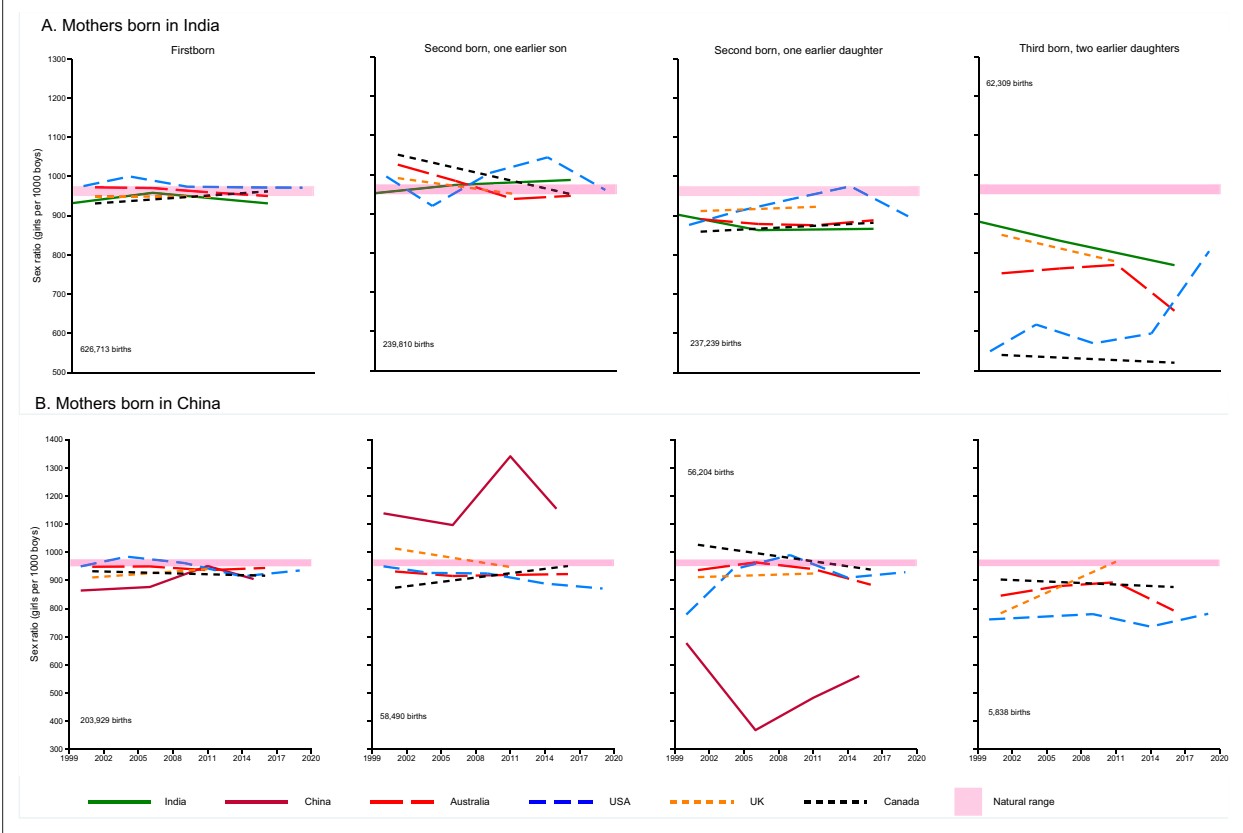

**Figure 1.** Conditional sex ratio (CSR) by birth order and country of birth. Second and third births in China indicate births in provinces or areas where more than one child was allowed. Solid lines represent births in India (green) and China (burgundy). Dash lines represent births among Indian and Chinese diasporas in Australia, Canada, UK, and US. CSRs shown are based on the total number of births (shown on each graph) for each birth order and its corresponding sex composition for all women of the specified background in each country. Deviations from the natural range are indicative of the difference in the observed CSRs from the baseline.

## Asian diasporas and domestic population

In Australia, Canada, the UK and the US, we contrasted the CSRs for second and third order births preceded by one or two earlier daughters among domestic-born mothers to the CSRs among diasporas of mothers born in China or India. (*Figure 2*). The CSRs for both birth orders were consistently within or close to the natural range of 950–975 for Australians, Canadians, the UK, and White Americans. In contrast, CSRs for the Chinese and Indian diasporas in each of these four countries deviated from the natural range for both birth orders with a more pronounced decline for third order births.

The CSRs for third order births with previous daughters among the Indian diaspora in all countries were substantially lower than for the Chinese diaspora, particularly in Canada where they also varied widely between second and third births for Indian women.

As a further check to see if the patterns among Indian-born mothers reflected subtle, but unmeasured biases in migration patterns, we did a further comparison of mothers born in India to those born in other South Asian countries in the UK, where migration from various South Asian countries is sufficiently common to assess such differences (*Figure 3*). Between the Indian diaspora and other Asian diasporas with a significant presence in the UK (*Figure 3*; *Supplementary file 1C*), the Indian diaspora had lower CSRs for both birth orders (second and third with previous daughters) compared to diasporas from Sri Lanka, Pakistan, and Bangladesh. CSRs for other Asian diasporas showed movement toward the natural range from 2001 to 2011 while CSRs for the Indian diaspora declined.

**Table 2.** Conditional sex ratio: second and third order births with one or two earlier daughters among Indian and Chinese women in the diaspora.

| Country (Year) | Birth order and sex of previous child | Indian | | | Chinese | | |
|---|---|---|---|---|---|---|---|
| Mother's ethnicity/ancestry | | Male | Female | Sex Ratio | Male | Female | Sex Ratio |
| Australia | Mother's country of birth | India | India | | China | China | |
| 2001 | Second born, one earlier daughter | 1267 | 1129 | 891 | 2017 | 1889 | 937 |
| | Third born, two earlier daughters | 155 | 116 | 748 | 214 | 181 | 846 |
| 2006 | Second born, one earlier daughter | 1592 | 1399 | 879 | 2428 | 2341 | 964 |
| | Third born, two earlier daughters | 167 | 127 | 760 | 241 | 212 | 880 |
| 2011 | Second born, one earlier daughter | 3061 | 2680 | 876 | 2904 | 2731 | 940 |
| | Third born, two earlier daughters | 261 | 201 | 770 | 282 | 252 | 894 |
| 2016 | Second born, one earlier daughter | 7419 | 6590 | 888 | 5271 | 4664 | 885 |
| | Third born, two earlier daughters | 596 | 389 | 653 | 421 | 334 | 793 |
| Canada | Mother's country of birth | India | India | | China | China | |
| 2001 | Second born, one earlier daughter | 618 | 531 | 859 | 451 | 463 | 1027 |
| | Third born, two earlier daughters | 137 | 74 | 540 | 62 | 56 | 903 |
| 2016 | Second born, one earlier daughter | 12275 | 10825 | 882 | 8370 | 7855 | 938 |
| | Third born, two earlier daughters | 2085 | 1085 | 520 | 810 | 710 | 877 |
| UK | Mother's country of birth | India | India | | China | China | |
| 2001 | Second born, one earlier daughter | 6049 | 5515 | 912 | 465 | 424 | 912 |
| | Third born, two earlier daughters | 1192 | 1009 | 846 | 60 | 47 | 783 |

*Table 2 continued on next page*

*Table 2 continued*

| Year | Mother's ethnicity/ancestry | Indian | | | Chinese | | |
|---|---|---|---|---|---|---|---|
| | Second born, one earlier daughter | 8378 | 7730 | **923** | 1366 | 1263 | **925** |
| 2011 | Third born, two earlier daughters | 1280 | 996 | **778** | 179 | 173 | **966** |
| | **Mother's country of birth** | | India | | | China | |
| **US** | | | | | | | |
| | Second born, one earlier daughter | 930 | 815 | **876** | 710 | 553 | **779** |
| 2000 | Third born, two earlier daughters | 182 | 100 | **549** | 109 | 83 | **761** |
| | Second born, one earlier daughter | 398 | 362 | **910** | 292 | 274 | **938** |
| 2004 | Third born, two earlier daughters | 68 | 42 | **618** | 42 | 42 | |
| | Second born, one earlier daughter | 1613 | 1521 | **943** | 1105 | 1094 | **990** |
| 2009 | Third born, two earlier daughters | 244 | 139 | **570** | 173 | 135 | **780** |
| | Second born, one earlier daughter | 1997 | 1947 | **975** | 1374 | 1250 | **910** |
| 2014 | Third born, two earlier daughters | 274 | 163 | **595** | 212 | 156 | **736** |
| | Second born, one earlier daughter | 2650 | 2381 | **898** | 1640 | 1524 | **929** |
| 2019 | Third born, two earlier daughters | 210 | 169 | **805** | 238 | 186 | **782** |

Example of CSR computation for Australia 2001 among Indian mother, parity 2: total births = 2396. Pf = 1129/2396=0.47, CSR = (0.47/(1–0.47)*1000=891). For parity 3: total = 271 pf = 116/271=0.43, CSR = (0.43/(1–0.43)*1000=748). Due to rounding of Pf value, CSR values may differ slightly.

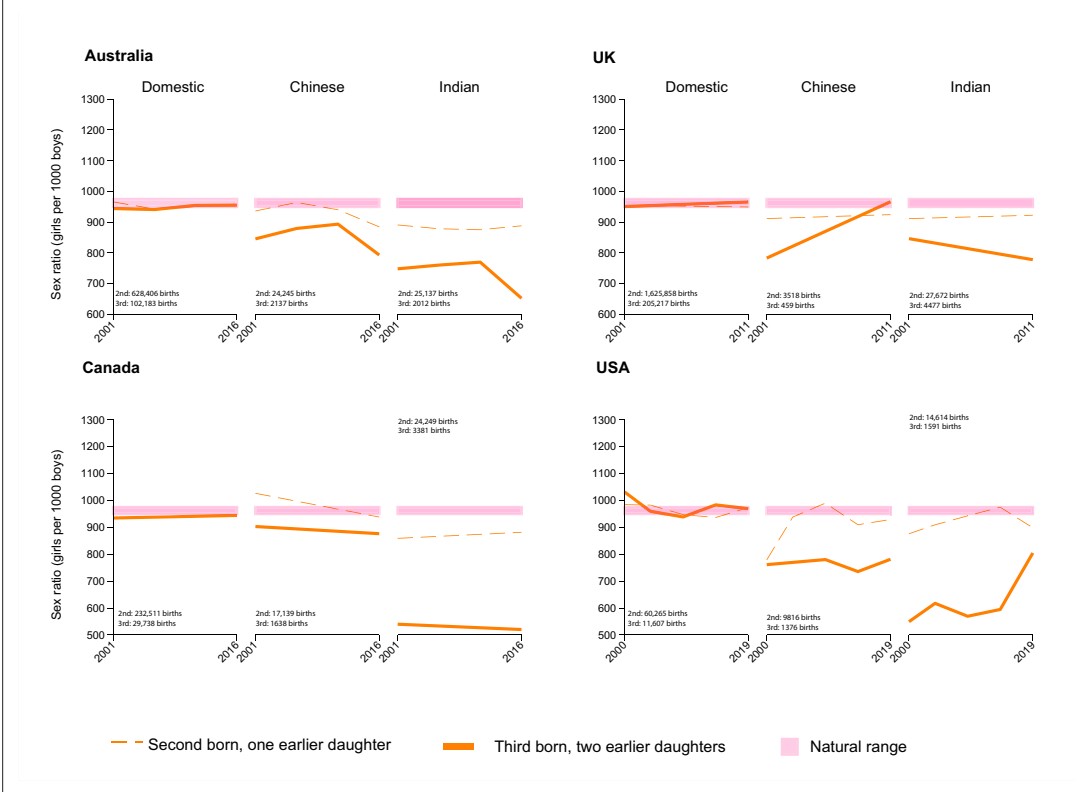

**Figure 2.** Conditional sex ratio (CSR) of second and third order births by mother's ethnicity and country of residence. CSRs shown are based on the total number of births (shown on each graph) for each birth order and its corresponding sex composition for all women of the specified background in each country.

## Discussion

We explored the patterns and trends of CSRs among home and diaspora populations of India and China. Depressed CSRs, suggesting missing girls at birth that arose almost all from selective abortion (*Almond et al., 2013*; *Almond and Edlund, 2008*; *Almond and Sun, 2017*; *Brar et al., 2017*; *Edvardsson et al., 2018*; *Edvardsson et al., 2021*; *Howell et al., 2017*; *Jha et al., 2011*; *Saikia et al., 2021*; *Urquia et al., 2016*), were observed for both Indian and Chinese populations, particularly for second and third order births following an earlier daughter or two earlier daughters. Among Indian women, CSRs were lower for third than second order births, and, surprisingly, were more pronounced in the diaspora population compared to India. Similarly, CSRs for third order births deviated more from the natural range than for second order births among women born in China. However, CSRs for second order births with either a previous son or daughter in China were more distorted compared to the Chinese diaspora. We showed evidence that selective abortion of male fetuses following a first-born boy occurs in China, along with the more commonly expected patterns of selective abortion of female fetuses following a first-born girl. We observed too few third order births in China to derive stable CSRs for comparison with the diaspora populations, among whom we documented low CSRs for third births. CSRs for second and third order births with earlier daughter(s) were lower for Indian than Chinese-born women in the diaspora.

Our findings were consistent with other studies showing similar CSR levels for India (*Jha et al., 2006*; *Jha et al., 2011*; *Saikia et al., 2021*). In Canada, the lowest CSRs were among Indians for third order births with two earlier daughters (*Almond et al., 2013*; *Brar et al., 2017*). This disproportion remained even without consideration of the sex composition of previous births, highlighting distorted sex ratios at third order births among Indian born mothers (*Urquia et al., 2016*). Sex ratios favouring boys were also observed for higher order births among Indian and Chinese born mothers in Australia (*Edvardsson et al., 2018*; *Edvardsson et al., 2021*) and the US (*Almond and Edlund, 2008*; *Almond and Sun, 2017*; *Howell et al., 2017*).

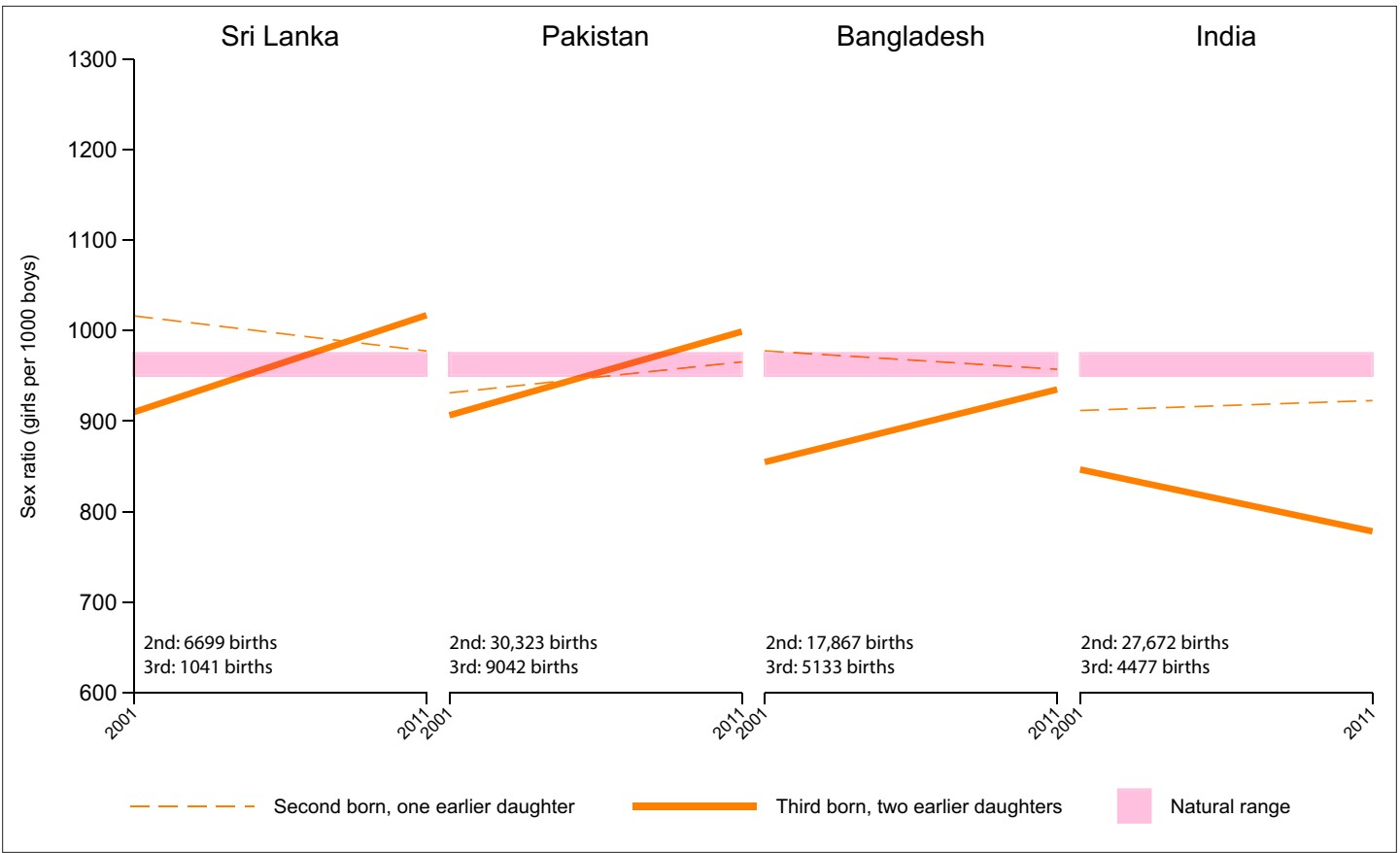

**Figure 3.** Conditional sex ratio (CSR) of second and third order births in the UK by mother's country of birth. CSRs shown are based on the total number of births (shown on each graph) for each birth order and its corresponding sex composition for all women of the specified background in each country.

In the three decades from 1987 to 2016, about 13.5 million girls were missing at birth India, worsening the already distorted population sex ratio (*Saikia et al., 2021*), while China is estimated to have 23.1 million missing girls at birth from 1970 to 2017 (*Chao et al., 2019*). These deviations, indicating prenatal sex selection (*Bongaarts and Guilmoto, 2015*; *Saikia et al., 2021*), remain a major concern as entrenched cultural preferences, ineffective government policies, falling fertility rates, and the availability of modern prenatal sex determination technology facilitate the practice of sex-selective abortions.

Until 2016, China's stringent one-child policy (1979–2015; with allowances for a second child under specific conditions) spurred sex selection even with policies prohibiting sex-selective abortions. The rise of CSRs from 1999 to 2015 for first births in China may be due to the relaxation of the one-child policy (*Ouyang, 2013*). However, China's one-child policy alone did not likely cause prenatal sex selection. Related factors such as delays in getting married and starting a family, changes in economic circumstances, and the demands of urbanization have also contributed to a decrease in fertility in China (*Almond et al., 2019*).

Numerous interventions to halt sex-selective abortions have had varying degrees of success in India and China (*Guo et al., 2016*). India banned the use of prenatal technologies in its 1994 Pre-Conception and Prenatal Diagnostic Techniques Act (*Tabaie, 2017*), expanded cash transfer schemes for the birth of girls and raised public awareness. China's large public gender equity campaigns (*Hesketh et al., 2011*; *Kumar and Sinha, 2019*), such as 'Care for Girls', monitored families suspected of practicing sex selection, especially those allowed to have a second child after an earlier daughter (*Kumar and Sinha, 2019*). Across Asia, regional and social variations drive the masculinized population (*Guilmoto, 2007*). Unlike India and China, Bangladesh and Pakistan had little evidence of selection (*Bongaarts and Guilmoto, 2015*) and our study showed their diaspora populations in the UK with rising CSRs towards the natural range in contrast to India. Notably, measures in the Republic of Korea effectively

brought the once skewed sex ratios close to the natural range (*Bongaarts and Guilmoto, 2015*; *Hesketh and Xing, 2006*; *Kumar and Sinha, 2019*; *UNFPA, 2020*).

The Indian diaspora had the most skewed CSR compared to the local population in India. This may be due to the high-quality healthcare services and easier access to abortion, particularly for immigrants with greater economic resources than in India. Canada, for instance, has a large share of foreign-born persons and has mostly unrestricted access to abortion (*Almond et al., 2013*). This may explain the low CSRs in Canada compared to the US, Australia, and the UK. For Chinese women, CSRs for second order births were more distorted in China than in the diaspora. This may indicate a weaker influence of cultural preference for sons for the latter while emphasizing the impact of China's one-child policy on fertility choices.

The stark consequence of prenatal sex selection plagues Asia with an excess of males who may never find brides. In China, unmarried sons may become burdens for their aging parents while families with daughters are incentivized to increase bride prices and select rich suitors. This could explain the spike in female births after a first son in China, for those permitted to have more than one child. People leaving China may do so to have more children, including girls (*Basten and Verropoulou, 2013*), and this might suggest that the observed CSRs at higher-order births among Chinese diaspora were closer to normal than might otherwise be expected. By contrast, Indian migrants are unlikely to migrate because of fertility restrictions, and the observed CSRs reflect the strong preference for boys even among migrants. As third order births are a relatively small contribution to overall birth totals in all countries and given that intermarriage rates between diaspora and non-diasporas are common (*Livingston and Brown, 2017*; *Yang and Bohm-Jordan, 2018*), Western countries do not face the profound demographic deficits of women as do India and China.

Our analyses were limited by the CHNS data, which were collected for select provinces in China. Given the varying local and provincial child policies during the study period, the estimated CSRs may not represent China at the national level. In addition, the skewed CSRs may be a result of underreporting of girls. However, the use of census and nationally representative survey data strengthen the findings of the study, and underreporting should not vary greatly by birth order. As discussed extensively earlier (*Jha et al., 2011*; *Saikia et al., 2021*), the CSR has some limitations, and is by definition a crude measure of selective abortion. Nonetheless it is unaffected by the factors that may reduce overall fertility. Finally, small numbers of observed births in some strata suggest caution in interpreting overall trends. The statistical uncertainty in any point estimate is wide, but the overall patterns over time should not be materially affected by the random variation in births per calendar year. We do not show confidence intervals for the CSRs, as the absolute numbers of births in the main comparisons are sufficiently large (*Table 1*) such that they would generate unjustifiably narrow confidence intervals. Moreover, the main uncertainty is not their point estimate, but their variation from the expected natural range, which is independent of birth order.

Overall, prenatal selection for sons prevails because of unchanging beliefs, cultural norms, most notably among Indians, and government policies - notably within China. Our data suggest that Indian diaspora have more depressed CSRs than do Chinese, largely reflecting strong boy preference that is portable across countries. Indeed, our study should help promote a debate as to why son preference leading to selective abortion of girls is far more widespread among Indians than Chinese. There is a need to raise public awareness about this issue and advocate for daughters in India and China where persistent deficits of girls aborted before birth may result in profound long-term social consequences.

## Data sharing

Data are publicly available except for Canada, UK, and Australia, which are available through the respective census bureaus. For China and India, data can be found on https://www.cpc.unc.edu/projects/china and https://dhsprogram.com/Countries/Country-Main.cfm?ctry_id=57&c=India after a free registration on both websites. A short proposal is also required to support data request from the DHS program.

## Acknowledgements

We thank Hellen Gelband for helpful suggestions and Lukasz Aleksandrowicz for earlier contributions to data collection.

# Additional information

### Competing interests

Prabhat Jha: Reviewing editor, eLife. The other author declares that no competing interests exist.

### Funding

| Funder | Grant reference number | Author |
|---|---|---|
| Canadian Institutes of Health Research | | Prabhat Jha |

The funders had no role in study design, data collection and interpretation, or the decision to submit the work for publication.

### Author contributions

Catherine Meh, Data curation, Formal analysis, Investigation, Visualization, Methodology, Writing – original draft, Writing – review and editing; Prabhat Jha, Conceptualization, Data curation, Formal analysis, Supervision, Funding acquisition, Investigation, Visualization, Methodology, Writing – original draft, Writing – review and editing

### Author ORCIDs

Catherine Meh  http://orcid.org/0000-0003-2476-8439
Prabhat Jha  http://orcid.org/0000-0001-7067-8341

### Ethics

No ethical approval was required for the study as no individual level identifiable patient data were used, only aggregated birth history data.

### Decision letter and Author response

Decision letter https://doi.org/10.7554/eLife.79853.sa1
Author response https://doi.org/10.7554/eLife.79853.sa2

# Additional files

### Supplementary files

• Supplementary file 1. Births and conditional sex ratios.
 (A) Births and conditional sex ratios (CSRs) from Australia Census 2001, 2006, 2011, 2016. (B) Births and CSRs from Canada Census 2001, 2016. (C) Births and CSRs from UK Census 2001, 2011. (D ) Births and CSRs from US Census and American Community Survey (ACS). (E) Births and CSRs for domestic population.

• MDAR checklist

### Data availability

Data are publicly available except for Canada, UK and Australia, which are available through the respective census bureaus. For China and India, data can be found on https://www.cpc.unc.edu/projects/china and https://dhsprogram.com/Countries/Country-Main.cfm?ctry_id=57&c=India after a free registration on both websites. A short proposal is also required to support data request from the DHS program.

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
