## [Editor Report]

This paper provides fundamental epidemiologic evidence on the abnormally skewed sex ratio at birth in Asia with expanded information on the Asian diaspora in OECD countries. India and China are two of the largest populations in the world that are most affected by a long history of strong son preference, aided by access to technology to detect the sex of the fetus and access to selective sex abortion. Both issues are important for maternal health, but also at the individual and family level in terms of kinship structure and the gender balance for future generations. The authors make a compelling case of skewed sex ratios favoring boys that persist among the Asian diaspora even under different social, cultural, or economic norms and constraints.

---

## [Decision Letter]

**Decision letter after peer review:**

Thank you for submitting your article "Sex ratio in the Asian diaspora: two decades of trend analyses in nationally representative census and survey data from Australia, Canada, the United Kingdom, and the United States" for consideration by *eLife*. Your article has been reviewed by 2 peer reviewers, and the evaluation was overseen by Eduarddo Franco as the Reviewing Editor and Senior Editor. The following individual involved in the review of your submission has agreed to reveal their identity: Patrick Gerland (Reviewer #1).

As is customary in *eLife*, the reviewers have discussed their critiques with one another and with the Reviewing and Senior Editors. The decision was reached by consensus. What follows below is a compilation of the essential and ancillary points provided by reviewers in their critiques and in their interaction post-review. Please submit a revised version that addresses these concerns directly. Although we expect that you will address these comments in your response letter, we also need to see the corresponding revision clearly marked in the text of the manuscript. Some of the reviewers' comments may seem to be simple queries or challenges that do not prompt revisions to the text. Please keep in mind, however, that readers may have the same perspective as the reviewers. Therefore, it is essential that you amend or expand the text to clarify the narrative accordingly.

Essential revisions:

1) Few small improvements for the plots to display the natural baseline would be welcome. Included plots help readers to synthesize a lot of detailed results for different groups of people, and it is a bit hard to keep track of the text alone, so the plots are welcome. The authors already discuss the statistical significance of the results, and it would probably be good to emphasize that the results are presented in terms of the number of cases used as denominators to compute rations, and the differences from the "natural" situation are statistically significant.

2) Consider the following suggestions:

– It is more conventional to measure sex ratios as # of males over females.

– It might be useful to benchmark CHNS for China against census data.

– Authors should not drop families with five or more children. Yes, these are few families, but one should retain all family sizes even when looking at say third parity gender. Dropping based on family size introduces a bias because family size is based on sibling sex composition, so, conditioning on family size introduces bias (Angrist Pischke, 2009). Granted, this bias will be small because there aren't many such families. but it's easy to do the correct way.

– While commonly assessed, the notion that the one-child policy cased sex selection is tenuous. fertility in China fell most during the 1970s, but sex selection did not increase. Fertility did not fall much in response to the introduction of the one-child policy in the early 1980s. This can be seen in aggregate fertility trends. Therefore, the mechanism behind the usual story doesn't really work. See Almond, Li, Zhang (2019). This isn't to say it's a settled issue, but rather that it is a contested one.

– Subsequent work has considered the health outcomes of the Asian Diaspora in the USA and evidence of gender bias. E.G.: https://pubmed.ncbi.nlm.nih.gov/33217631/ and https://www.researchgate.net/publication/341832217_Disability_among_children_of_immigrants_from_India_and_China_Is_there_excess_disability_among_girls

*Reviewer #1 (Recommendations for the authors):*

Meh and Jha investigate the persistence of son preference, especially for high birth orders (i.e., 2nd or 3rd child), among the first and second generations of Asian migrants living in Australia, Canada, the UK, and the USA. The authors provide a concise, but rich set of summary statistics and an effective set of plots to highlight the systematic deviations from the norms, the similarities in levels and trends across countries and groups, and persistent differences.

Strengths: Excellent paper overall. The analysis and presentation are clear, the data used by the author permit the required level of disaggregation to perform the more in-depth analysis required. The outcome of interest (conditional sex ratio) is not as well known as the simpler/usual sex ratio at birth, but this more specific indicator offers the authors explain a better alternative since it takes into account the sex of previous births, and is more effective to analyze the sex ratio of higher parities.

Weaknesses: No significant weaknesses. Only a few clarifications might be needed to help readers such as to include in the footnote an example of how the CSR gets computed for 1 country-year and parity 2 and 3.

Line 162 and appendix table: clarify for analysis and appendix table that CSR is computed only if enough number of births. Specify the minimum threshold used (n=60?) and justify why not a higher number as the denominator.

Figure 1 is already with a lot of info but it would be helpful to also include the “natural” range as a baseline.

Figures 2 and 3: hard to see the "natural" range in the plots and legends.

*Reviewer #2 (Recommendations for the authors):*

Among sex ratio studies, the paper is unusually comprehensive in studying both data from primary/origin countries and the diaspora as well. This is a competent and timely study that documents some interesting heterogeneities in sex selection patterns across countries, races, and family composition.

---

## [Author Response]

Essential revisions:1) Few small improvements for the plots to display the natural baseline would be welcome. Included plots help readers to synthesize a lot of detailed results for different groups of people, and it is a bit hard to keep track of the text alone, so the plots are welcome. The authors already discuss the statistical significance of the results, and it would probably be good to emphasize that the results are presented in terms of the number of cases used as denominators to compute rations, and the differences from the "natural" situation are statistically significant.

We have updated the figures to better show the natural sex ratio range and have noted the significance of the results in the notes section of the figures.

2) Consider the following suggestions:– It is more conventional to measure sex ratios as # of males over females.

In order to highlight missing female births, a female to male ratio is preferable. We have clarified this choice – line 131.

– It might be useful to benchmark CHNS for China against census data.

We have addressed this in our data source section of the manuscript line 87.

– Authors should not drop families with five or more children. Yes, these are few families, but one should retain all family sizes even when looking at say third parity gender. Dropping based on family size introduces a bias because family size is based on sibling sex composition, so, conditioning on family size introduces bias (Angrist Pischke, 2009). Granted, this bias will be small because there aren't many such families. but it's easy to do the correct way.

We have already excluded large families, as they do not show significant evidence of prenatal sex selection. The preconditions of prenatal sex selection include the preference to bear a son, access to prenatal services and the need for small family size. Some large families may show evidence of postnatal bias towards boys and neglect of girls, which may lead to excess female mortality in infancy and childhood. However, child mortality rates are low in western countries. India and China have seen marked improvements with declines in child mortality. Hence, sex selection is largely at the prenatal phase and among couples desiring a small family. Thus, we focus our analyses on these families with four children or less where there is sex selection particularly among second and third order births.

– While commonly assessed, the notion that the one-child policy cased sex selection is tenuous. fertility in China fell most during the 1970s, but sex selection did not increase. Fertility did not fall much in response to the introduction of the one-child policy in the early 1980s. This can be seen in aggregate fertility trends. Therefore, the mechanism behind the usual story doesn't really work. See Almond, Li, Zhang (2019). This isn't to say it's a settled issue, but rather that it is a contested one.

We agree that China’s one child policy did not cause sex selection and advance that other factors like delays in getting married and starting a family, changes in economic circumstances, and the demands of urbanization contributed to a decrease in fertility in China. Instead, our study results suggest that the one child policy had an impact on sex selection, which was already in existence before the introduction of the policy. In addition, the one child policy was not uniformly enforced across China. Thus, the impact of this policy would vary. Prior to the introduction of prenatal techniques, postnatal sex selection was prevalent and was reflected in the neglect of girls and high female infant and child mortality in places where son preference prevailed. With the subsequent introduction and access to prenatal ultrasound technology, abortion services, and a desire for a small family size, prenatal sex selection increased in the form of sex selective abortions of females. The impact of prenatal sex selection is demonstrable in the excess male population in China. We have added some of these points to the discussion and now cite the Almond publication.

– Subsequent work has considered the health outcomes of the Asian Diaspora in the USA and evidence of gender bias. E.G.: https://pubmed.ncbi.nlm.nih.gov/33217631/ and https://www.researchgate.net/publication/341832217_Disability_among_children_of_immigrants_from_India_and_China_Is_there_excess_disability_among_girls

Thank you for the references. We agree that there may be evidence of postnatal sex selection, which is seen in the disparity between the care of female versus male children. This is, nonetheless, beyond the scope of this analysis, which seeks to document evidence of missing girls at birth.

Reviewer #1 (Recommendations for the authors):Meh and Jha investigate the persistence of son preference, especially for high birth orders (i.e., 2nd or 3rd child), among the first and second generations of Asian migrants living in Australia, Canada, the UK, and the USA. The authors provide a concise, but rich set of summary statistics and an effective set of plots to highlight the systematic deviations from the norms, the similarities in levels and trends across countries and groups, and persistent differences.Strengths: Excellent paper overall. The analysis and presentation are clear, the data used by the author permit the required level of disaggregation to perform the more in-depth analysis required. The outcome of interest (conditional sex ratio) is not as well known as the simpler/usual sex ratio at birth, but this more specific indicator offers the authors explain a better alternative since it takes into account the sex of previous births, and is more effective to analyze the sex ratio of higher parities.Weaknesses: No significant weaknesses. Only a few clarifications might be needed to help readers such as to include in the footnote an example of how the CSR gets computed for 1 country-year and parity 2 and 3.

CSR formula is shown line 128. We have included an example of CSR computation in the footnote of Table 2.

Line 162 and appendix table: clarify for analysis and appendix table that CSR is computed only if enough number of births. Specify the minimum threshold used (n=60?), and justify why not a higher number as the denominator.

We specify that we derive CSRs for each birth order with a minimum total of 100 births (female/male) in the methods section line 139.

Figure 1 is already with a lot of info but it would be helpful to also include the “natural” range as a baseline.

The color of the natural range bar has been darkened for clarity.

Figures 2 and 3: hard to see the "natural" range in the plots and legends.

The color of the natural range bar has been darkened for clarity.